# How Early-Life Programming During Embryogenesis Imprints Cellular Memory

**DOI:** 10.3390/ijms27010163

**Published:** 2025-12-23

**Authors:** Norermi Firzana Alfian, Kei Uechi, Yoshiya Morishita, Kaname Sato, Maruhashi Yui, Jannatul Ferdous Jharna, Md. Wasim Bari, Shiori Ishiyama, Kazuki Mochizuki, Satoshi Kishigami

**Affiliations:** 1Department of Integrated Applied Life Science, Integrated Graduate School of Medicine, Engineering, and Agricultural Sciences, University of Yamanashi, Kofu 400-0016, Japan; g24dib02@yamanashi.ac.jp (N.F.A.); g24dib01@yamanashi.ac.jp (K.U.); g24dib05@yamanashi.ac.jp (Y.M.); g24lb011@yamanashi.ac.jp (K.S.); g24lb031@yamanashi.ac.jp (M.Y.); janna.just.2016@gmail.com (J.F.J.); 2Faculty of Life and Environmental Sciences, University of Yamanashi, Kofu 400-8510, Japan; wasim.bc36@yahoo.com (M.W.B.); i-shiori@yamanashi.ac.jp (S.I.); mochizukik@yamanashi.ac.jp (K.M.)

**Keywords:** cellular memory, epigenetic inheritance, embryogenesis, DOHaD

## Abstract

Cellular memory, or epigenetic memory, represents the capacity for cells to retain information beyond the underlying DNA sequence. This heritable characteristic is primarily governed by epigenetic mechanisms which enable cells to maintain specialized characteristics across divisions. This persistent cellular state is essential for fundamental biological processes, such as maintaining tissue identity and facilitating cell differentiation, especially embryonic cells. Early-stage perturbations such as assisted reproductive technologies (ART) and nutritional stress links embryonic exposures to adult health and disease within the Developmental Origins of Health and Disease (DOHaD) framework. Crucially, memory established during early embryogenesis links these epigenetic modifications to adult long-term phenotypes related to metabolic disorders. These modifications—including DNA methylation, histone modifications, and non-coding RNAs—support cellular memory transmission across cell divisions, and in certain organisms, can be transmitted across generations without alterations to the DNA sequence. This review synthesizes recent advances in epigenetic pathways that mediate cellular memory, highlights critical preimplantation windows of vulnerability and outlines gaps necessary for mammalian developing interventions that safeguard future generations.

## 1. Introduction

The developmental origins of health and disease (DOHaD) theory proposes that environmental factors and exposures during critical periods of early development even before organogenesis such as during preconception, can have lasting effects on an individual’s physiology and metabolism, influencing the risk of developing chronic diseases in later life. When we integrate with the classical “all-or-none” theory, where it suggests that these exposures may result in either embryonic death or complete recovery, it could be interpreted that certain sublethal exposures may induce molecular or cellular alterations that allows the embryo to adapt which will persist within the cells across divisions [1]. Thus, DOHaD explores how early-life exposures, such as nutrition and stress, can induce lasting changes in cellular function and gene expression. This concept refers to a term “cellular memory”, which can be interpreted in multiple ways but in the context of development it encompasses mechanisms that enable cells to retain information influenced by prior environmental conditions, allowing them to pass through cellular divisions and differentiation without genetic alterations [2]. The recent interest in cellular memory, especially in embryology, centers around how epigenetic alterations or modifications from environmental factors contribute to the developmental trajectories. Researchers explore this concept through various terms, such as the inherited metabolic cellular memory together with the epigenetic memory [3]. This factor is particularly significant in embryo development, as early environmental exposures can greatly influence developmental processes and later health outcomes based on the theory that inherited metabolic cellular memory consists of regulatory adaptations from environmental variables especially diet [4]. Studying cell memory in embryo development is crucial for understanding how environmental factors can influence the long-term health and disease predisposition. This is possible from the embryo adaptability based on environmental responsiveness or developmental plasticity to ensure survivability, but the alterations can also lead to maladaptive programming later in life. It is still unclear if the adaptative responses lead to improved metabolic regulatory output or cause intergenerational diseases with no cure. Further understanding the regulatory alterations could be vital for comprehending how non-communicable metabolic diseases are formed.

Cellular memory established during embryogenesis is essential because it reveals the epigenetic marks that could be linked to adult phenotypes such as psychological disturbances, impaired spatial memory, insulin resistance, type-2-diabetes, hypertension, cardiovascular disease, and fertility issues [5]. It provides the mechanistic bridge between early environmental inputs and later health outcomes, making its study indispensable for understanding and preventing diseases. The inheritance of these environmentally induced changes, known as transgenerational epigenetic inheritance, is a key area of study but highly debated [6]. Transgenerational studies show that exposure to pollutants, a high-fat diet, or diabetes can propagate reproductive deficits and metabolic dysfunction for two or more generations [7]; this is from “transgenerational epigenetic inheritance”, which refers to the transmission of gene expression changes across generations without altering the DNA sequence, suggesting that DOHaD-related phenotypes can be reproduced through altered developmental processes based on the cellular memory of a single cell, namely the germ cell. These effects are mediated through alterations of the germ cell function, established during critical developmental windows; however, due to the extensive epigenetic reprogramming during early embryogenesis these interpretations should be critically viewed as this process exerts certain limitations towards stable transmission of epigenetic markers across generations. Research conducted on paternal diet showed significant influence on offspring metabolism through germ cells such as embryo and sperm parameters, creating an “inherited metabolic cellular memory” that can affect multiple generations [8]. This can suggest that not only the exposed individual, but also future generations may be affected by ancestral exposures by indirectly modifying the threshold risk towards developing certain metabolic conditions. Deciphering these mechanisms is vital for developing strategies to prevent and mitigate diseases linked to early-life environmental exposures.

This literature review consolidates recent findings from various studies on cellular memory, aiming to clarify the underlying mechanisms, with a focus on epigenetic regulation and the effects of external stimuli, particularly their implications in lifelong development. Among these, we have focused our discussion on the long-term metabolic disease consequences resulting from exposures during early embryogenesis.

## 2. Background

Historically, embryology has studied how embryos develop over a course of time, while developmental biology, which emerged in the 1960s, shifted toward more molecular and genetic-driven processes [9]. It began with the introduction of epigenetic by Conrad Waddington in 1940 that defined it as the complex developmental processes connecting genes and the environment [10,11]. Later, in 1958 David Nanney suggested referring to epigenetics as the control mechanisms of gene expression that produce differentiated cell phenotypes originating from the same genotype but are perpetuated through cellular division [12]. The idea of differentiated cells maintaining their phenotype was later integrated as the basis of cellular memory.

The DOHaD hypothesis originated from David Barker, who observed the relation of intrauterine conditions that permanently altered fetal structure and metabolism, which later increases the risk towards adult diseases such as heart disease and metabolic disorders [13]. From the concept of cellular memory, this suggests that early-life environmental exposures can reprogram the health trajectory solely due to the fetal adaptative drive for survival that can be detrimental later in life [12,14,15]. Modern understanding in DOHaD, recognizes that epigenetic mechanisms such as DNA methylation and histone modifications are the few mechanistic outcomes from external stimuli that allows cells to retain the “memory” without DNA sequence alterations or mutation [16]. These alterations are not only maintained through cell divisions and differentiation but can also be transmitted to the offspring, influencing the disease risk of offsprings from exposed individuals. This can be seen from the studies of historical famines that provide valuable evidence on this transgenerational epigenetic effect of malnutrition [17,18]. The integration of cellular memory, epigenetics, and environmental influence provides the molecular basis for the long-term health consequences outlined by the DOHaD theory.

## 3. Fundamental Principles of Cellular Memory: Epigenetic and Metabolic Regulation

Cellular memory refers to the phenomenon whereby transient environmental stimuli or metabolic states experienced by a cell are retained over long periods without altering the underlying DNA sequence, thereby exerting sustained effects on subsequent cellular functions and phenotypes. Early embryos exhibit extremely high sensitivity to environmental cues, and alterations in maternal nutritional status, inflammation, oxidative stress, and metabolic balance during pregnancy (the embryonic developmental period) are thought to reprogram the embryonic epigenome, establishing transcriptional programs that persist after birth [19,20]. As shown in Figure 1, this section outlines the core concepts underlying such cellular memory, focusing on two major mechanisms: epigenetic memory—comprising DNA methylation, histone modifications, and chromatin architecture—and metabolic memory, whereby changes in metabolic states influence epigenomic regulation.

### 3.1. Epigenetic Memory

DNA methylation is a representative epigenetic mechanism that confers long-term stability to transcriptional regulation. In particular, methylation of CpG islands promotes transcriptional repression through the formation of heterochromatin. The maintenance of methylation patterns following DNA replication is mediated by DNA methyltransferase 1 (DNMT1), which recognizes hemi-methylated DNA and faithfully copies the parental methylation pattern onto the daughter strand, ensuring long-term epigenomic stability. In contrast, de novo methylation during development is catalyzed by DNA methyltransferase 3A (DNMT3A) and DNA methyltransferase 3B (DNMT3B), which exhibit distinct region-specific functions. DNMT3A primarily introduces new methylation marks into promoter regions, especially CpG islands near transcription start sites, thereby acting as a “transcriptional off-switch” that stably establishes the ON/OFF state of specific genes. DNMT3B, by contrast, preferentially targets gene bodies rather than promoters and functions to repress transcriptional elongation [21]. Through the coordinated interplay of DNMT1-mediated maintenance methylation, DNMT3A- and DNMT3B-mediated de novo methylation, and multiple layers of histone modification, a robust and stable framework of cellular memory is established during development and in response to environmental stimuli.

Histone modifications constitute another major component of cellular memory. Histones undergo diverse chemical modifications—including acetylation, methylation, phosphorylation, and ubiquitination—and the combinatorial patterns of these marks, known as the histone code, precisely regulate chromatin structure and transcription. Among these modifications, histone H3 lysine 4 trimethylation (H3K4me3) serves as a canonical transcriptional activation mark, whereas histone H3 lysine 9 trimethylation (H3K9me3), histone H3 lysine 27 trimethylation (H3K27me3), and histone H3 lysine 36 trimethylation (H3K36me3) function as transcriptional repression marks. Histone acetylation induces an open chromatin conformation and facilitates transcriptional activation, whereas histone methylation, despite causing no net charge change, promotes stable chromatin states through the recruitment of specific reader proteins, thereby contributing to the maintenance of developmental transcriptional programs. Acetylated histones are also recognized by dedicated reader proteins that modulate gene expression. These modifications are dynamically regulated by writer enzymes, reader proteins, and eraser enzymes, enabling both rapid adaptability to environmental stimuli and long-lasting cellular memory across cell divisions.

Chromatin architecture, shaped by the integration of DNA methylation and histone modifications, can be broadly classified into euchromatin and heterochromatin. Euchromatin exhibits an open configuration with enriched histone acetylation and demethylation and permits active gene transcription required for development and environmental responses. In contrast, heterochromatin is characterized by dense DNA methylation and repressive histone modifications, maintaining long-term transcriptional silencing. Because chromatin structure is relatively stable across cell divisions, transient stimuli encountered during embryonic development can be converted into long-lasting alterations in gene expression programs, forming “structural memory,” which provides a physical basis for the persistence of cellular memory [22].

### 3.2. Metabolic Memory

Enzymes involved in epigenomic regulation utilize a range of metabolite-derived cofactors and substrates—including S-adenosylmethionine (SAM), acetyl–CoA, α-ketoglutarate (αKG), nicotinamide adenine dinucleotide (NAD^+^), and flavin adenine dinucleotide (FAD)—produced through cellular metabolism. Consequently, changes in cellular metabolic states exert direct effects on epigenomic modifications. Maternal macronutrient balance during pregnancy, one-carbon metabolism–related nutrients such as folate, vitamin B6, vitamin B12, methionine, and choline, and the intake of B-vitamins that influence NAD^+^ and FAD levels alter embryonic concentrations of SAM, acetyl-CoA, αKG, NAD^+^, and related metabolites. These fluctuations subsequently modulate DNA methylation and histone modification states. SAM is essential for both DNA and histone methylation [23], and acetyl–CoA serves as the principal donor for histone acetylation [24]. αKG is indispensable for the activity of Ten-Eleven Translocation (TET) enzymes and JmjC domain–containing histone demethylases [25]. NAD^+^ is required for the activity of the Sirtuin family of deacetylases; inflammation and obesity reduce the NAD^+^/NADH ratio, diminish Sirtuin activity, increase histone acetylation, and disrupt gene expression [26]. FAD acts as a cofactor for lysine-specific demethylase 1 (LSD1) and depends on vitamin B2 availability [27]. Thus, shifts in cellular metabolite levels are converted into epigenomic alterations that persist beyond transient metabolic changes and become fixed as “metabolic cellular memory,” a stable form of cellular memory maintained across cell divisions.

### 3.3. Mechanisms by Which Maternal and Embryonic Nutritional–Metabolic Environments Establish Cellular Memory

Taken together, these metabolism–epigenome interactions clearly indicate that the nutritional and metabolic environment during pregnancy and embryonic development plays a decisive role in shaping cellular memory in early embryos. The maternal protein–fat–carbohydrate (PFC) balance, one-carbon metabolism nutrients (including folate, vitamins B6 and B12, choline, and methionine), intake of vitamins such as niacin and riboflavin, and maternal inflammation, oxidative stress, and metabolic abnormalities can modulate embryonic levels of SAM, acetyl–CoA, αKG, NAD^+^, and related metabolites, thereby altering the embryonic epigenome. These epigenomic modifications may not represent transient responses but could instead become stably fixed as cellular memory, persisting across cell divisions and influencing postnatal metabolic regulation and susceptibility to lifestyle-related diseases. This framework suggests a molecular explanation for the central concept of the Developmental Origins of Health and Disease (DOHaD), which posits that environmental conditions during pregnancy critically determine long-term health outcomes.

## 4. External Influences During Embryogenesis

### 4.1. Nutritional and Metabolic Influences

Embryonic development is highly sensitive to nutritional and metabolic inputs from both the maternal environment and artificial culture systems. As the DOHaD paradigm proposes that maternal nutrition and environmental exposures can leave a lasting cellular memory in the embryo, influencing organ development and long-term susceptibility to disease. Environmental factors present at conception exert profound influences on mammalian development and can alter the long-term physiological health of offspring after adaptative period to ensure survivability [28,29,30,31,32].

Evidence from multiple species underscores the importance of early nutritional conditions and environmental exposures as summarized in Table 1. For example, Valazquez et al. found that both under- and over-nutrition trigger molecular adaptations in preimplantation embryos, potentially compromising embryo quality [33]. Similarly, Cai et al. demonstrate that nutritional imbalances can cause epigenetic modification disorders affecting embryo development [19]. Rodent studies further demonstrate that maternal protein restriction during preimplantation development is associated with abnormal postnatal growth and greater susceptibility to chronic disease later in life [34,35,36,37]. Poor maternal nutrition has been linked to increased risk of hypertension, metabolic dysfunction, and behavioral deficits in adulthood [38,39,40,41].

In addition to maternal nutrition, in vitro culture media serve as the sole nutrient source for embryos generated via assisted reproductive technologies (ART) [42]. Early studies highlighted the requirement for amino acids in culture systems to support cleavage, blastocyst formation, and viability [43,44,45]. Furthermore, it has recently been revealed that the nutritional environment provided by culture media induces epigenetic changes in embryos [46,47,48]. Bovine serum albumin (BSA), a standard component of conventional media, contributes to osmotic regulation, pH buffering, lipid solubilization, and small-molecule transport. Perturbations in nutrient supply, such as BSA-free or low-protein conditions, reduced inner cell mass (ICM) and trophectoderm (TE) cell numbers, thereby altering the ICM/TE ratio and compromising blastocyst quality [49].

Embryos exhibit remarkable metabolic plasticity, allowing them to survive suboptimal environments; however, these short-term adaptations often incur long-term liabilities [50]. For instance, exposure to elevated glucose levels can induce aberrant glycolytic flux and the accumulation of reactive oxygen species (ROS), leading to mitochondrial dysfunction [51]. Concurrently, oxidative stress can dysregulate redox-sensitive signaling pathways, forcing further recalibrations of energy metabolism [52,53]. Critically, these compensatory adjustments can become embedded as a form of epigenetic or metabolic cellular memory [54]. However, parental abnormalities such as obesity can alter the transcriptome of the offspring’s early embryos, thereby contributing to the transmission of disease risk to the first filial (F1) generation [55,56]. Chen et al. further confirmed this by reporting that hyperglycemia can dysregulate the epigenome during early fetal development, potentially impairing organ development [57]. Petropoulos et al. also provided cross-species evidence showing gestational diabetes triggers widespread DNA methylation changes that affect metabolic pathways [58]. Nutrient driven epigenetic modifications key mechanisms include one carbon metabolism regulating DNA methylation have found to be affecting stem cell differentiation and potential long-term health consequences from periconceptual conditions [59,60]. Peral-Sanchez et al. reviewed how maternal undernutrition, high-fat diets, low-protein regimens, and excess folic acid supplementation during embryo culture or in utero remodel DNA methylation, histone modifications and non-coding RNA (ncRNA) patterns, predisposing children to obesity and cardiovascular disease later in life [61].

Overall, nutritional conditions decisively alter embryonic development, converging to establish metabolic and epigenetic imprints that persist throughout life [62,63]. Understanding these pathways is essential not only for optimizing reproductive outcomes but also for mitigating intergenerational risk of chronic disease through tailored nutritional interventions. While the current evidence is compelling, further comprehensive research are required to elucidate the complex interactions between nutrition, metabolism, and the epigenetic programming during early development.

### 4.2. In Vitro Physical and Chemical Exposures

The DOHaD hypothesis indicates that environmental conditions during fetal development significantly impacts adult health status and the risk of lifestyle-related diseases [64,65]. Meanwhile, assisted reproductive technologies (ART), such as in vitro fertilization (IVF) embryo culture and cryopreservation, increase the embryo’s exposure to external environments. Consequently, evidence in recent years suggests that offsprings conceived through ART may experience various effects in adulthood [66,67], although some reports suggest context-dependent differences [68].

Insights from assisted reproductive technologies have further illuminated the sensitivity of preimplantation development. Embryos cultured in vitro encounter conditions distinct from the maternal reproductive tract, and randomized controlled trials have demonstrated that culture-medium composition can affect birthweight, intrauterine growth, and neonatal outcomes even when genetic background is held constant [69,70,71]. When embryos and sperm cells are exposed to the in vitro environment, fluctuations in pH and temperature, along with oxidative stress, degrade embryo quality [72,73,74]. Damage to embryos during culture duration due to the gap with the in vivo environment leads to abnormal phenotypes in offspring, and offspring derived from IVF embryos have an increased risk of lifestyle-related diseases [75]. The effects of vitamin B supplementation on in vivo and in vitro fertilized embryos showed an inverse response accompanied with the increased risk towards glucose intolerance in early adulthood suggesting that in vitro fertilization alters the embryo adaptative response to nutrition supplementation compared to in vivo fertilization [76]. Single-embryo transcriptomic studies show that different media induce stage-specific transcriptional divergence, indicating that preimplantation sensitivity is tied to developmental milestones rather than being uniformly distributed across time [77]. It was further highlighted how oxidative stress during in vitro manipulation can impair sperm function [78]. Sperm oxidative stress impairs embryo development through dose-dependent effects on motility and oxidative status as demonstrated in experiments using hydrogen peroxide on bull sperm [79]. In vitro manipulation of spermatozoa can lead to damage, particularly to DNA integrity, which is crucial for fertilization and genetic transmission [80]. Interpretation of ART-associated long-term health outcomes, however, requires careful consideration of both treatment-related exposures and underlying parental characteristics [81].

Cryopreservation introduces additional stress. Freezing and thawing can damage cell membranes and genetic material [82], and while vitrification—a method of ultra-rapid freezing—is now the mainstream technique, vitrified embryos exhibit reduced cell numbers and lower development rates compared to fresh embryos [83]. In addition, cryoprotectants contained in the freezing and thawing solutions used for vitrification have been reported to possess significant toxicity [84,85,86]. Moreover, offspring derived from vitrified embryos not only exhibit different birth weights compared to those from fresh embryos, but significant weight gain abnormalities and impaired glucose and lipid metabolism have been reported in adults derived from frozen embryos [87,88,89]. Some transcriptomic studies in vitrified blastocyst reported changes in expression of genes for translation-related pathways, which raises the possibility that vitrification might affect ribosome biogenesis at the blastocyst stage, potentially affecting long-term protein synthesis and health outcomes [90].

Additionally, intracytoplasmic sperm injection (ICSI), which involves directly injecting sperm into the oocyte, is effective for improving fertilization rates and treating male infertility. However, concerns exist regarding potential damage to the egg and sperm caused by the physical injection process [91]. Recently, however, piezo ICSI has been clinically applied. This technique reduces damage to the egg by applying a vibration (piezo pulse) to the cell membrane using a pipette with a flat tip [92]. Furthermore, the use of chorionic villus sampling (CVS) to screen embryos for chromosomal abnormalities is increasing, aiming to reduce miscarriage rates. While this is expected to enable the high-precision selection of healthier embryos, risks such as low birth weight and small-for-gestational-age offsprings have been demonstrated [93,94].

Environmental factors beyond ART also affect embryogenesis. Other physical and chemical factors encountered in vitro may disrupt embryonic development [95]. Exposure to electromagnetic radiation (EMR) at 2.4 GHz has been shown to induce oxidative stress and apoptosis in chick embryos and neuronal cell lines, suggesting that external physical stimuli can perturb cellular integrity and potentially influence developmental outcomes [96]. Chemical exposures exert similar effects: bisphenol A (BPA) during spermatogenesis alters sperm genetic and epigenetic information, leading to embryonic apoptosis and impaired survival [97]; cadmium exposure dramatically reduces blastocyst formation and sperm motility [98]; and toxicants can disrupt epigenetic marks at key developmental loci, potentially affecting offspring phenotypes across generations [99].

As ART continues to evolve with novel techniques, embryos are increasingly exposed to diverse stresses. Evidence indicates that such exposures may have long-term consequences for offspring health, yet mechanistic insights remain limited due to the relatively recent implementation of these technologies. To improve ART safety and safeguard the health of future generations, continued investigation into the molecular and epigenetic impacts of ART procedures is imperative.

### 4.3. Windows of Sensitivity

As mentioned earlier, the DOHaD framework posits that adult health is shaped not only by inherited genetics but also by environmental conditions experienced during early development. Although fetal and neonatal periods have historically been considered primary windows of vulnerability, mounting evidence extends this sensitivity back to the preimplantation stage—the short but highly dynamic interval between fertilization and implantation—during which the embryo undergoes profound molecular and cellular transitions [100]. Furthermore, it has been reported that culture conditions themselves, such as in vitro culture, also affect gene expression in early embryos [101], and differences in culture medium composition may influence developmental timing and the expression profile.

A central event during this period is the maternal-to-zygotic transition (MZT), wherein developmental control shifts from maternally deposited transcripts to newly activated zygotic transcription. In mammals, major zygotic genome activation (ZGA) occurs at the two-cell stage in mice and between the four- and eight-cell stages in humans, accompanied by substantial chromatin remodelling and transcriptomic reconfiguration [102,103,104]. These transitions make the embryo especially sensitive to even transient nutritional, metabolic, or iatrogenic disturbances, which may be recorded as persistent molecular modifications. Shortly thereafter, compaction and polarization initiate the first lineage-allocation events, segregating the inner cell mass (ICM) from the trophectoderm (TE). Because the TE forms the placenta, disruptions to its specification can disproportionately influence fetal development and long-term metabolic health [105]. The placenta, derived from the trophectoderm, acts as a critical mediator through which early environmental signals modulate fetal and postnatal physiology. Early perturbations can remodel placental architecture, nutrient transport, vascularization, and endocrine function, shape fetal growth trajectories and influencing susceptibility to cardiometabolic diseases later in life [106,107].

Animal models provide mechanistic clarity regarding how transient preimplantation exposures become biologically embedded. Mouse embryos briefly exposed to α-minimum essential medium (αMEM) give rise to adults exhibiting post-prandial hyperglycaemia and heightened inflammatory gene expression [108]. More prolonged exposure induces kidney-disease–like pathology, which can be ameliorated through dietary interventions such as barley supplementation [109]. These models reinforce the dual themes of vulnerability and reversibility within early developmental programming.

Perinatal outcomes observed in ART-conceived children further underscore the intersection between early environmental exposures and lifelong health risks, supporting the broader DOHaD principle that the earliest days of life exert disproportionate and durable effects on long-term physiology [110]. Together, human and animal studies converge on a unified concept: the preimplantation embryo contains narrow, milestone-anchored windows of heightened sensitivity during which environmental exposures can impart enduring biological effects. Clinically, this emphasizes the need for embryo-culture systems that faithfully emulate in vivo reproductive-tract conditions. Scientifically, it highlights the importance of linking preimplantation exposures with long-term follow-up, dissecting placental mechanisms of mediation, and exploring nutritional and metabolic strategies capable of mitigating adverse trajectories. Optimizing this earliest developmental window represents a promising frontier for chronic-disease prevention and intergenerational health.

**Table 1 ijms-27-00163-t001:** Summarized research in relation to embryonic cellular memory affecting epigenetic memory and onset metabolic diseases.

Author	Year	Species	Findings	Exposure	Ref.
Kwong et al.	2000	Rat	Blastocyst defects and hypertension	Nutrition	[28]
Watkins et al.	2010	Mouse	Effects vascular metabolism		[35]
Eckert et al.	2012		Blastocyst metabolic reprogramming		[37]
Seki et al.	2017		Hepatic hypermethylation and metabolic changes		[111]
Upadhyaya et al.	2017		Cardiac histone modifications		[112]
Lessard et al.	2019		Multigenerational male semen defects		[8]
Crisóstomo et al.	2021		Inherited altered metabolites		[17]
Pepin et al.	2022		Inherited metabolic dysfunction and histone methylation		[113]
Tang et al.	2023		Methylation and metabolism		[114]
Whatley et al.	2023		Metabolic alteration and altered histone acetylation		[115]
Whatley et al.	2024		Altered histone acetylation		[116]
Tomar et al.	2024		Metabolic inheritance by sperm mitochondrial RNAs		[117]
Zhu et al.	2024		Disrupts development and metabolism		[56]
Alfian et al.	2025		Altered heterochromatin methylation		[76]
Desmet et al.	2016	Bovine	Epigenetic and transcriptomic changes		[118]
Fernández-González et al.	2004	Mouse	Alters genes and behavior	Culture media	[38]
Watkins et al.	2007		Increases systolic blood pressure		[40]
de Lima et al.	2020		Embryo metabolic adaptation		[51]
Ishiyama et al.	2021		Hyperglycemia and higher inflammatory genes		[108]
Ishiyama et al.	2021		Diabetic kidney disease		[109]
Whatley et al.	2022		Altered development and metabolism		[119]
Sato et al.	2025		Improve glucose intolerance and in vitro culture effects		[120]
da Fonseca Junior et al.	2023	Bovine	Adaptive metabolic and epigenetic		[121]
Dumoulin et al.	2010	Human	Influences offspring birthweight		[69]
Nelissen et al.	2013		Influences fetal growth		[70]
Ducreux et al.	2023		Altered gene expressions		[77]
Khosla et al.	2001	Mouse	Altered imprinted gene expression	Culture conditions	[47]
Ecker et al.	2004		Long-term behavioral effects		[7]
Mahsoudi et al.	2007		Transgenerational effects		[39]
Banrezes et al.	2011		Affects adult body weight		[41]
Uysal et al.	2022		Altered DNA methylation		[46]
Moriyama et al.	2022		Altered gene expression and metabolites		[72]
Jharna et al.	2025		Alters metabolism		[49]
Donjacour et al.	2014	Mouse	Alters glucose metabolism	IVF	[75]
Bai et al.	2022		Heterochromatin alterations and placental defects		[122]
Bari et al.	2023		Increased birth rates		[123]
Lee et al.	2025		Proteomic and metabolic response		[4]
Cui et al.	2020	Human	Increased metabolic dysfunction risk	ART exposure	[124]
Huang et al.	2021		Lower DNA methylation		[125]
Ling et al.	2009	Mouse	Impacts developmental competence	Cryopreservation	[83]
Qin et al.	2021		Abnormal glucose metabolism		[89]
Chen et al.	2024		Affects metabolism		[88]
Lee et al.	2024		Metabolic and gene expression alterations		[90]
Pavlinkova et al.	2017	Mouse	Male subfertility inherited	Diabetes exposure	[18]
Chen, B.	2022	Mouse	Transmission of glucose intolerance		[126]
Petropoulos et al.	2015	Human	Offspring DNA methylation changes		[58]
Rechavi et al.	2014	*C. elegans*	Small-RNA-based transgenerational inheritance	Starvation	[127]
Deena et al.	2025	Chicken	ROS generation and apoptosis-induced	Electromagnetic radiation	[96]

## 5. Mechanisms of Cellular Memory

Early embryonic development is orchestrated by precise regulated spatiotemporal gene expression and metabolic systems [128]. Concurrently, the embryo undergoes global epigenetic reprogramming. Near-complete erasure of DNA methylation is followed by lineage-specific re-establishment, and wide-ranging histone modification resetting occurs in parallel. These processes provide the embryo with exceptional developmental plasticity yet simultaneously create acute vulnerability to environmental perturbations [129].

Metabolism in early embryos is characterized by stringent nutritional requirements, but as development progresses, they acquire metabolic plasticity, accompanied by flexible changes in the expression of metabolic enzymes [130]. Metabolic pathways intersect closely with this epigenetic resetting. One-carbon metabolism—integrating dietary folate, methionine, and B vitamins—supplies essential methyl donors for chromatin methylation during the preimplantation period. Experimental perturbation of these pathways alters DNA methylation patterns and compromises developmental programming, underscoring that metabolic inputs directly influence the fidelity of early embryonic epigenetic re-establishment [131,132]. Mitochondria provide an additional regulatory layer by generating metabolites necessary for chromatin-modifying enzymes, thereby linking cellular bioenergetics with epigenomic programming. Altered mitochondrial dynamics during this window reduce developmental competence and increase susceptibility to later-life disease [133]. In addition, changes in gene expression—particularly in epigenome-modifying enzymes and in the levels of their metabolic substrates and cofactors—can reshape the epigenetic landscape of early embryos [121]. These changes may accumulate throughout development and ultimately result in functional abnormalities and disease phenotypes in adulthood.

### 5.1. DNA Methylation

Epigenetic modifications, including DNA methylation are thought to provide a molecular cellular memory of the past conditions encountered during early life events. This process is tightly controlled by programmed waves of demethylation and subsequent re-establishment of methylation patterns following fertilization and during primordial germ cell development. Key aspects linking DNA methylation to embryonic memory include imprinting control regions (ICRs) that are protected from genome-wide demethylation waves that occur shortly after fertilization, thereby preserving a parent-of-origin-specific epigenetic mark or memory [134]. In adult cells, hypomethylation DNA at specific enhancer sites act as a stable cellular memory of tissue-specific developmental enhancer activity from the embryonic stage, potentially enabling the recovery of those embryonic genes [135].

Research confirms that DNA methylation is a critical mechanism for establishing and maintaining cellular memory during embryonic development, linking early programming to later life outcomes. As previously discussed, nutritional exposures programs long-term epigenetic modifications by DNA methylation. Exposure to high-fat diet in utero showed association with increased incidence of cardiovascular disease, diabetes and metabolic syndrome later in life. This was further investigated causes DNA hypermethylation, which is associated with gene expression changes in liver of exposed offspring potentially contributing to programmed development of metabolic diseases later in life [111]. Studies also found gene methylation, namely *Irs2* and *Map2k4*, in the offspring liver, which in turn predisposes the offspring to diabetes later in life [136]. Non-esterified fatty acid concentrations causes high variation in DNA methylation however main gene networks affects were related to lipid and carbohydrate metabolism, cell death, immune response and metabolic response [118]. Li et al., specifically showed that maternal conditions like obesity can trigger DNA methylation alterations that predispose offspring to metabolic diseases such as diabetes and fatty liver disease [137]. Furthermore, maternal metabolic disorders, including gestational diabetes, have been demonstrated in multiple studies to induce changes in the expression of DNA methylation–related enzymes in oocytes [138]. In the context of somatic cell nuclear transfer (SCNT) embryo models, research has found that SCNT-derived embryos are able to maintain the DNA methylation patterns throughout development, which supports the consistent role of DNA methylation in embryonic cellular memory [139].

DNA methylation plays essential roles not only in transcriptional repression but also in the regulation of imprinted genes and X-chromosome inactivation (XCI) during development. Gametes are highly methylated, but undergo extensive demethylation after fertilization, followed by remethylation after implantation. Diabetes has been shown to reduce the expression of TET3, a DNA demethylation enzyme, in oocytes, thereby altering the methylome of early embryos and contributing to the transmission of impaired glucose tolerance to the F1 generation [126]. DNA methylation could be transmitted between generations in certain mammals, however the functional importance and whether it carries information across generations are unclear. For instance, young transposable elements, due to their evolutionary age, may retain DNA methylation levels during early embryonic development, impacting embryonic gene expression [140]. Despite that, other underlying factors such as genetic mutations or indirect mechanisms should be considered in transgenerational transmission.

### 5.2. Histone Modifications

Histone modifications, including methylation, acetylation, ubiquitination, phosphorylation, and lactylation, play crucial roles in regulating gene expression by modulating chromatin structure and accessibility. As reported by Matoba et al., histone methylation can act as a determining factor on embryonic development capabilities presented by observing SCNT embryo models [141]. The four core histones assemble into an octamer, around which DNA is wrapped to form the nucleosome—the fundamental unit of chromatin.

Nutritional status and culture conditions are closely linked to epigenomic modifications. For instance, maternal ketogenic diet feeding or in vitro exposure to ketone bodies alters histone 3 lysine 27 acetylation (H3K27ac) levels in trophectoderm (TE) cells of blastocysts [115,116,119]. A study into alterations in genome-wide histone modifications in neonatal hearts of rats exposed to maternal diabetes and a high-fat diet. Many of the genes were associated with the metabolic process, particularly with “positive regulation of cholesterol biosynthesis”. These data reveal that maternal diet changes the histone signature in offspring in utero, suggesting a fuel-mediated epigenetic reprogramming [112]. Furthermore, in the extraembryonic ectoderm of in vitro–fertilized embryos, aberrant H3K4me3 patterns have been associated with the ectopic activation of epiblast-specific genes [122].

Nutritional exposure not limited to maternal but also the paternal effect was further explored. Obesity in males alters sperms H3K4me3 patterns, particularly at genes involved in metabolic, inflammatory, and developmental pathways, and these changes are transmitted to embryos and the placenta. Experimental overexpression of the histone demethylase enzyme lysine (K)-specific demethylase 1A (KDM1A) in the germline further disrupts H3K4 methylation, leading to offspring metabolic dysfunction. These findings suggest that sperm H3K4me3 acts as a metabolic sensor linking paternal diet to offspring phenotypes via the placenta [113]. Study in worms showed transmission of sperm H3K27me3 patterns to embryos and for important development consequences of paternal marking which demonstrate the paternal inheritance involvement in embryogenesis [142].

One of the pivotal studies in this field investigated the role of Polycomb group proteins and the histone modification H3K27me3 in maintaining the silencing of developmental regulators. This research revealed that H3K27me3 is extensively present in oocytes, where it is associated with regions lacking transcription and DNA methylation. Upon fertilization, a notable loss of promoter H3K27me3 at critical developmental genes such as Hox genes was observed, indicating a resetting of epigenetic memory. This resetting process highlights the plasticity of the epigenome during gametogenesis and early development, suggesting that cellular memory is not static but rather subject to reprogramming [143]. Studies using *C. elegans* also demonstrated that H3K27me and Polycomb repressive complex 2 (PRC2) each contribute to epigenetically transmitting the memory of repression across generations and during development [144].

### 5.3. Role of Non-Coding RNAs in Maintaining Memory

Embryos exposed to such maternal environmental changes exhibit modified gene expression patterns, particularly within metabolic pathways [145]. Several studies report that small non-coding RNAs (ncRNAs) act as vectors of transgenerational inheritance following nutritional stress. Non-coding RNAs (ncRNAs) are a diverse group of RNA molecules that do not encode proteins but exert a wide range of regulatory functions. For instance, microRNAs (miRNAs) bind to target RNAs to reduce their stability or translation efficiency, thereby repressing gene expression at the post-transcriptional level. They also contribute to the degradation of maternal RNAs during zygotic genome activation (ZGA) in the transition from fertilized egg to early embryo [146]. Long non-coding RNAs (lncRNAs) regulate gene expression through multiple mechanisms, including transcriptional and post-transcriptional regulation as well as chromatin modification [147]. During embryonic development, the lncRNA transcribed from the Xist gene on the X chromosome plays a central role in X-chromosome inactivation (XCI) [148,149].

In recent years, ncRNAs have attracted considerable attention as potential molecular mediators of the transmission of parental environmental information to the next generation. Sperm-derived ncRNAs have been reported to influence embryonic development and the metabolic and behavioural phenotypes of offspring [150]. Indeed, paternal metabolic disorders such as obesity can alter the miRNA profile in sperm, which in turn affects the metabolic phenotype of the progeny [151]. In mice, sperm or testes RNA—including microRNAs and non-coding RNAs from obese fathers was reported to induce metabolic phenotypes in naïve embryos, with microRNA-19b specifically shown to recapitulate diet-induced metabolic alterations [152]. Sperm-derived microRNAs were also implicated in the transmission of behavioral and metabolic phenotypes following early life stress in rodents, with microinjection experiments reported to demonstrate causality [153]. Such environmentally induced changes in paternal ncRNAs are transmitted to early embryos, where they perturb transcriptional and metabolic programs, thereby potentially contributing to postnatal phenotypic outcomes [117]. On the other hand, maternal environments also influence early embryos. Oviductal and uterine fluids of pregnant mice contain abundant tRNA-derived small RNAs (tsRNAs) and rRNA-derived small RNAs (rsRNAs)**,** whose expression levels are altered in response to a high-fat diet. In *Caenorhabditis elegans*, starvation induces small RNAs that are inherited for at least three generations [127].

### 5.4. Nuclear Structure and Chromatin Remodeling Complexes

In mammals, the nucleolus during early embryogenesis appears as nucleolar precursor bodies (NPBs), which are compact structures lacking transcriptional activity [154]. As development progresses, NPBs gradually transition into mature, somatic-type nucleoli at the morula stage, coinciding with the onset of ribosomal RNA transcription [155]. From the transition of embryonic NPBs to somatic tripartite nucleolus, this transition involves dramatic structural, yet some nucleolar proteins remain anchored in the core, potentially acting as a molecular memory that safeguards essential nucleolar functions during reorganization [156,157]. However, the function of this inactive NPBs is still unknown, but its morphology acts as a marker to predict the offspring viability [123,158].

The nucleolus is tightly associated with heterochromatin, it was reported that without nucleolus the zygotic chromatin is highly disorganized [159,160]. Moreover, the maternal nucleolus plays a critical role in maintaining centromeric satellite sequences, whereas paternal chromatin undergoes extensive remodelling after fertilization due to protamine-to-histone replacement [161,162]. Before ZGA, there is no transcriptional activity, maternal inherited proteins and RNAs which are utilized to support embryos until their own transcription activity starts. During ribosome biogenesis, the pre-rRNA (ribosomal RNA) transcript undergoes a series of precisely coordinated cleavage and modifications events. One of the most critical events in the maturation process is the removal of 5′ ETS (5′ external transcribed spacer). When this cleavage step is impaired, it leads to the accumulation of unprocessed ribosomal RNA (rRNA) intermediates, disrupting the normal assembly of ribosomal subunits. Because rRNA processing and ribosome assembly occur within the nucleolus these defects cause significant alterations in nucleolar organization such as changes in size and morphology [163]. Even as the nucleolus undergoes dramatic reorganization, the persistence of certain core proteins may serve as a molecular memory, guiding the re-establishment of nucleolar core function to maintain cellular homeostasis along with the integration with global chromatin architecture [164,165].

Unlike membrane-bound organelles, the nucleolus is a membrane-less condensate formed by liquid–liquid phase separation (LLPS). This property makes its morphology highly sensitive to environmental cues such as pH, ionic strength, protein concentration, and post-translational modifications including phosphorylation, acetylation, methylation and deamination. In addition, molecular factors such as non-coding RNAs, particularly Alu RNA, have emerged as key regulators of nucleolar architecture [166,167]. Alu RNA interacts closely with nucleolar proteins to stabilize nucleolar subdomains, preserve nucleolar integrity, and prevent dispersion under stress conditions. Mechanistically, this regulation supports the assembly and activity of rRNA transcriptional machinery, ensuring efficient ribosome biogenesis [168]. In the context of embryonic development, such stabilization of nucleolar organization by Alu RNA may be especially critical during the cleavage and blastocyst stages, when rapid cell divisions and high translational demand require robust rRNA synthesis to support lineage specification and developmental progression [169,170]. The dynamic, liquid-like nature of nucleoli allows them to rapidly reorganize in response to stress or metabolic changes.

In early embryos, the embryonic nucleolar plasticity is tightly coupled to chromatin remodelling events, which establish transcriptional competence and higher-order nuclear organization (Figure 2). Nucleolar organization also influences genome–nucleolus interactions, helping to define repressive chromatin domains and contributing to the establishment of embryonic cell memory. Thus, the transition from NPBs to mature nucleoli represents a developmental checkpoint, where structural plasticity of the nucleolus and large-scale chromatin reorganization converge to guide embryonic fate decisions [171,172].

### 5.5. Ribosome and rRNA Processing

The nucleolus serves as the central hub of ribosome biogenesis, where rRNA transcription, processing and ribonucleoprotein assembly are tightly coordinated within a dynamic, phase separated microenvironment [173]. Nucleolar morphology such as shape, size and number are directly linked to rRNA transcription [173,174]. During early embryonic development, low levels of ribosomal DNA (rDNA) methylation promote high rDNA transcription, which in turn enhances nucleolar activity—supporting the elevated demands for ribosome biogenesis required for rapid cell growth and differentiation. The association between lower rDNA copy number and adult obesity has been well documented in both humans and mice [175,176]. Furthermore, both decreased rDNA copy number and hypermethylation of rDNA are linked to obesity in adulthood, suggesting that rDNA integrity serves as a form of developmental memory connecting early-life nutritional environments to long-term metabolic outcomes [177]. This concept of ‘functional rDNA memory’ proposes that the regulatory state of rDNA during embryogenesis, rather than copy number alone—shapes nucleolar architecture and establishes lasting imprints that influence adult physiology and susceptibility to disease.

Building on this framework, early rRNA transcription dynamics appear to play a similarly critical role. Under various stress conditions, cells may initiate rRNA transcription; however, the subsequent processing of these transcripts often becomes inefficient. In mice, the ribosomal DNA transcription is initiated at the 2-cell stage, while an abrupt shift towards ribosomal DNA replication begins at the 4-cell stage [178,179]. In *Drosophila* embryos, rDNA transcription is detectable even before the nucleolus is morphologically visible, indicating that the transcriptional activity can itself drive nucleolar assembly rather than relying on pre-existing nucleolar structures [180]. Although total rDNA copy number does not directly correlate with nascent rRNA transcription rates, the transcriptional output from active rDNA repeats plays a critical structural role in nucleolar organization [181,182]. Nascent rRNA molecules act as molecular surfactants that limit excessive growth of fibrillar centers (FC), demonstrating that nucleolar integrity depends not merely on rDNA abundance but on the regulation of transcriptional activity and its architectural consequences [183,184].

Pharmacologic inhibition of rRNA synthesis further illustrates the sensitivity of nucleolar organization to transcriptional dynamics. Such inhibition can trigger nucleolar stress, with the specific nature of the response depending on the molecular target of each compound. In such cases, unprocessed rRNA can persist within the nucleolar compartment, and upon stress resolution, ribosome biogenesis can resume [185,186]. However, whether the recovered ribosomes are structurally or functionally altered remains unclear. CX-5461, for example, irreversibly inhibits RNA polymerase I initiation at rDNA promoters without broadly affecting RNA polymerase II, thereby imposing specific disruption of ribosomal biogenesis and nucleolar phase-separation properties [187,188]. In embryos, treatment with CX-5461 does not necessarily prevent development; however, the long-term consequences of this early nucleolar stress on adult physiology and the potential emergence of specific phenotypes remain unexplored.

Together, these observations suggest that the dynamic regulation of rRNA transcription and nucleolar architecture may contribute to the establishment of embryonic epigenetic memory (Figure 3). Elucidating how these diverse stressors shape nucleolar dynamics will be essential for understanding how early-life environments imprint long-lasting effects on physiology and disease susceptibility.

### 5.6. Autophagy

Autophagy is rapidly activated after fertilization in mammals, with its expression markedly upregulated during the early stages of mouse embryogenesis. In mammals, autophagy is an early embryonic nutrient supply system involved in early embryonic development, implantation, and pregnancy maintenance. Recent studies have found that autophagy imbalance in placental tissue plays a key role in the occurrence and development of pregnancy complications, such as gestational hypertension, gestational obesity, premature birth, miscarriage, and intrauterine growth restriction [189]. Genetic studies provide compelling evidence for the indispensable role of autophagy in early embryos. Embryos deficient in autophagy, such as those lacking the autophagy-related gene Atg5, fail to develop beyond the 4- to 8-cell stage and undergo preimplantation lethality [190]. During the nutrient-limited preimplantation period, autophagy appears to support embryonic development and cell differentiation by degrading oocyte-derived proteins and cytoplasmic components to secure essential nutrients.

A study on the association between maternal hyperglycaemia and postnatal high-fat diet exposure compromises metabolic parameters and hepatic autophagy in female offspring showed reduced phosphorylated AMP-activated protein kinase (p-AMPK) and autophagy-related protein light chain, LC3-II/LC3-I ratio, the presence of steatosis, oxidative stress, and inflammation later on in adulthood. The reduction in autophagy, stimulated by high-fat diet, may be of vital importance for susceptibility to metabolic dysfunction-associated fatty liver disease induced by maternal diabetes [191]. This suggests that alterations in nutrient availability or autophagy-mediated degradation during the preimplantation window may influence long-term developmental outcomes [192]. We examined the long-term effects of inhibiting autophagy in preimplantation embryos using chloroquine (CQ), an autophagy inhibitor, and found that CQ mitigates the long-term effects of in vitro culture in mouse embryos [120]. This suggests that altered autophagy activity in vitro culture contributes to adverse long-term outcomes. Numerous large-scale meta-analyses and population cohort studies on human assisted reproductive technologies have documented various, yet generally modest differences in metabolic traits—fasting glucose, insulin, lipid profiles, blood pressure, and fat percentage between ART and naturally conceived (NC) children in the literature, many of which diminish with age [66,124,125,193,194]. Studies have also shown that IVF children have lower birth weights and higher rates of congenital anomalies compared to NC children [195]. These issues may stem from differences in the culture media used.

While further investigation into culture media composition and new embryo culture technologies is essential, modulating autophagy during preimplantation may offer a promising strategy to mitigate ART-associated risks. In summary, approaching nutritional status and the long-term effects of in vitro culture from the perspective of the autophagy pathway may be key as a novel in vitro culture technology.

## 6. Transgenerational Cellular Memory

Nutritional and environmental exposure can perturb embryonic metabolism and gene expression, and the long-term consequences of culture have been widely explored. Embryonic cellular memory—primarily through epigenetic programming, has profound lifelong implications for disease susceptibility (Figure 4) [196]. One major consequence is the increased risk of metabolic and chronic disease in adulthood following exposure to adverse intrauterine environments. Conditions such as maternal malnutrition or obesity can induce “fetal metabolic programming”, predisposing offspring increased susceptibility to disorders including obesity, insulin resistance, and cardiovascular diseases later in life [197,198].

Transgenerational cellular memory provides an important extension in embryonic cellular memory, particularly in understanding how environmental exposures may influence not only the individuals but also subsequent generations. Some studies have reported the parental exposures have caused developmental impacts on their offsprings; however, these effects remain highly debated due to limited understanding on how this transmission occurs as well as the mechanisms in transmission despite the epigenetic reprogramming that occurs during early embryonic development eliminating inheritance marks [199]. Evidence from certain studies showing that early-life nutritional status and parental high-fat diets can induce heritable epigenetic modifications, making offspring more susceptible to conditions like obesity and diabetes [114,200]. This could suggest the possibility that the epigenetic alterations could be due to an indirect cellular memory bias established during early development, when interacting with individual postnatal experiences produces similar tendencies towards disrupted metabolisms. This process of “transgenerational epigenetic programming” can involve changes in the epigenome of germ cells, affecting phenotypes like diabetes, insulin resistance, hypertension, and brain development in future generations should be considered within a broader framework of cellular memory [201]. To eliminate the possibility of genetic mutations that could impose in certain organisms, consideration surrounding this epigenetic programming inheritance are essential when determining the transmission mechanism [202]. The influence of the environmental factors on epigenetic cellular memory has been further explored in the context of morphine exposure. Research has demonstrated that morphine can induce stable epigenetic changes in mouse embryonic stem cells (mESCs), which persist even after the withdrawal of the drug. This finding emphasizes the potential for environmental exposures to create long-lasting transcriptional and metabolic changes in gene expression [203]. 

Collectively, these findings highlight transgenerational embryonic cellular memory as an important layer in the mechanistic framework linking early environmental exposures to long-term physiological outcomes by modifying baseline disease susceptibility, not only for the individual but potentially across multiple generations. Rather than focusing transgenerational cellular memory representing a fixed epigenetic inheritance, another view is by the transmission of modulated cellular metabolic and developmental threshold responses to the next generation.

## 7. Conclusions

In conclusion, the exploration of embryo cellular memory encompasses a multifaceted approach that integrates epigenetic regulation, early environmental influences, and developmental context interact to shape long-term health trajectories. The interplay between genetic, epigenetic, and environmental factors during the earliest stages of life establishes molecular imprints at the cellular level, with implications for health and disease in later life. As research continues to unravel the complexities of embryonic cellular memory, it is essential to consider both biological and environmental contexts to foster a comprehensive understanding of embryonic development. This holistic perspective will not only enhance scientific knowledge but also inform clinical practices aimed at improving reproductive outcomes.

Despite major advances, a significant gap lies in isolating the precise effects of individual assisted reproductive technology interventions particularly in vitro embryo culture, on epigenetic reprogramming and health outcomes. Addressing these limitations through standardized experimental designs will be essential in enabling precise evidence-based understanding on developmental programming. Although considerable attention has focused on DNA methylation in relation to ART, the contributions of cellular memory discussed remain comparatively underexplored and warrant deeper mechanistic investigation.

Furthermore, there is a need for more extensive studies on the transgenerational epigenetic effects of ART and embryo culture, as well as the long-term health consequences for offspring. Current research often relies on animal models and a limited number of imprinted genes, indicating the need for broader epigenetic analyses using human samples using high-throughput methods. Broader, high-resolution epigenomic analyses using human samples are urgently needed to correlate its direct causal impact on humans, warranting careful considerations for clinical applications.

Gaps also persist in understanding the paternal contribution to embryonic cellular memory. How environmental factors influence the sperm epigenome, and how these changes are transmitted across generations to affect offspring phenotypes remains largely unresolved. Clearer mechanistic insights into sperm-mediated retention and transmission of epigenetic marks—including DNA methylation, chromatin modifications, and non-coding RNAs—are essential for elucidating the paternal origins of developmental programming.

Finally, a major conceptual challenge lies in defining the direct relation between specific epigenetic marks and gene expression. Many studies report methylation differences without corresponding transcriptional changes, underscoring the need for integrative analyses that connect epigenetic variation to functional outcomes. Continued research in this direction will be crucial for understanding how early-life environments shape cellular memory and, ultimately, organismal health across the lifespan and generations. These studies will pave the way for the development of diagnostic methods to decode the environmental history of cells in individuals and predict future diseases, as well as for strategies to prevent both future disease and the transgenerational inheritance of adverse effects, ultimately aiming to promote health throughout life.

## Figures and Tables

**Figure 1 ijms-27-00163-f001:**
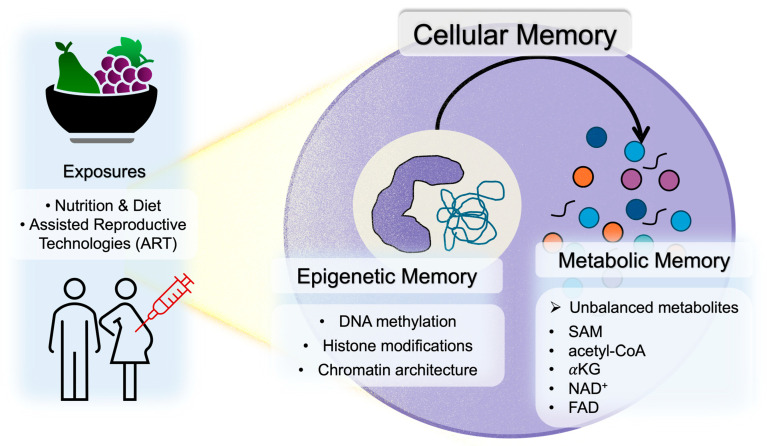
Overview of cellular memory encompassing epigenetic and metabolic memory.

**Figure 2 ijms-27-00163-f002:**
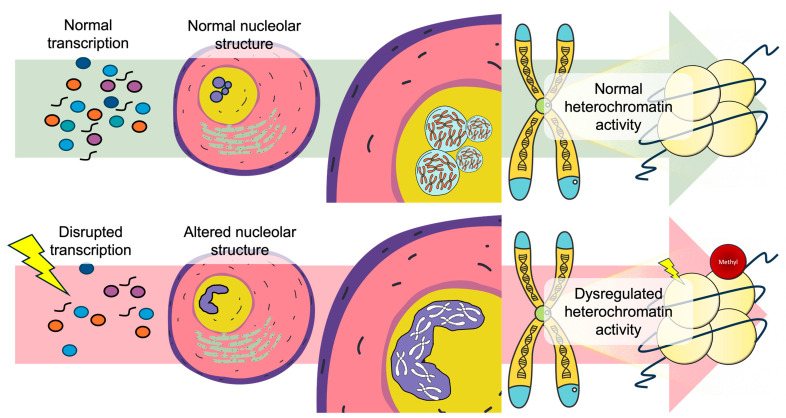
Nucleolar structural integrity as a regulator linking transcriptional homeostasis to chromatin architectural stability. Under normal transcription conditions (**top**), proper ribonucleoprotein assembly and nuclear organization supports normal chromatin activity. In contrast (**bottom**), environmental exposures with the ability to influence transcription disruption cause nucleolar structural alterations leading to dysregulated chromatin interactions.

**Figure 3 ijms-27-00163-f003:**
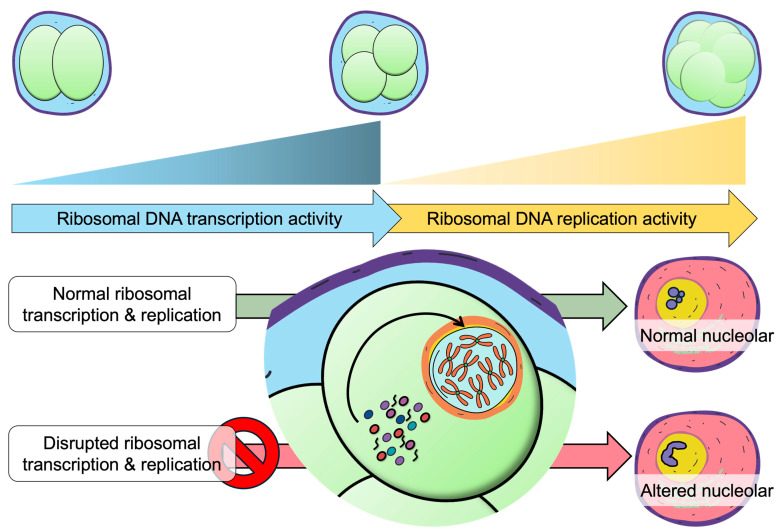
Ribosomal DNA transcription-replication dynamics correlation with nucleolar structural maturation.

**Figure 4 ijms-27-00163-f004:**
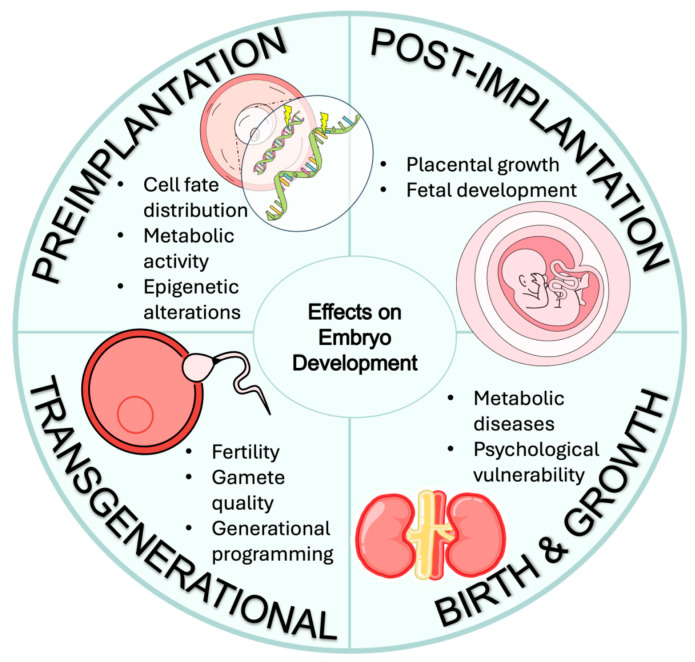
A summary of multistage potential effects on embryo development caused by nutrition or environmental exposure.

## Data Availability

No new data were created or analyzed in this study. Data sharing is not applicable to this article.

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
