# Peer review of "How Early-Life Programming During Embryogenesis Imprints Cellular Memory"

_ijms, 2025, doi:10.3390/ijms27010163_

Round 1
Reviewer 1 Report
Comments and Suggestions for Authors
The presented manuscript is an interesting review summarizing new, often controversial data and concepts. Unfortunately, the authors do not make efforts to present also conflicting data and concepts, including theories that have current and long-lasting status of scientific consensus. In recent decades, the enthusiasm accompanying the fast progress of epigenetics led to the publication of many reports about trans-generational non-genetic inheritance without the needed scrutiny, and a revival of Lamarckian views on heredity. In addition to specific cases where such claims have been explicitly disproved (see references below), the concept suffers from the general logical issue of any theory of inheritance of acquired traits: the hidden assumption that the conditions in which the organism’s ancestors developed and lived have a paramount importance, while the conditions in which the organism itself develops and lives have little importance, or none at all. Because the authors are mostly interested in the long-lasting effects (into adulthood) of factors acting on the early embryo, I do not even understand why they put so much stress on transgenerational epigenetic inheritance – if anything, if such a phenomenon exists, it would determine the fate of embryos by factors that had acted generations ago, and thus diminish the importance of factors acting on the actual embryo here and now, which are the main topic of the review.
Lines 21-22: There are yet no compelling reasons to suppose that such transmission exists in humans (or other mammals), see references below.
Lines 32, 205-206, 329-332: These statements contradicts the “all-or-none” theory of early embryonic exposure to damaging factors (e.g. Adam MP. The all-or-none phenomenon revisited. Birth Defects Res A Clin Mol Teratol. 2012; 94(8):664-9. doi: 10.1002/bdra.23029), which is the long-lasting scientific consensus and should be referenced even if the authors disagree with it and cite some data supporting the opposite concept.
Line 38-39, 60-71, 97-101, 435-436, 645 ff.: Transgenerational epigenetic inheritance is highly controversial, particularly in mammals (e.g. Heard E, Martienssen RA. Transgenerational epigenetic inheritance: myths and mechanisms. Cell. 2014; 157 (1): 95-109. doi: 10.1016/j.cell.2014.02.045; Horsthemke B. A critical view on transgenerational epigenetic inheritance in humans. Nat Commun. 2018; 9(1):2973. doi: 10.1038/s41467-018-05445-5; Bird A. Transgenerational epigenetic inheritance: a critical perspective. Front Epigenet Epigenom. 2024; 2: 1434253. doi: 10.3389/freae.2024.1434253). “In mammals efficient reprogramming occurs in the early embryo and in the germ line... These two rounds of epigenetic erasure leave little chance for inheritance of epigenetic marks, whether programmed, accidental or environmentally induced” (Heard and Martienssen, 2014). Data initially considered as supporting transgenerational epigenetic inheritance have turned out to be based on mutations: “Several studies… have reported the co-segregation of an abnormal DNA methylation pattern… with a rare disease in two or more generations of certain families. In these cases, the abnormal DNA methylation of the gene under investigation was linked to a mutation in a neighboring gene that removed the transcription termination signal... As a consequence, transcription from this gene extended into the gene under investigation, causing abnormal promoter methylation and gene silencing” (Horsthemke, 2018).
Lines 193-196: The authors should either provide a reference, or frame the statement as a hypothesis still to be tested.
Lines 260-261: Other studies do not find significant differences between children conceived naturally and those conceived using assisted reproduction (e.g. Luke B, Brown MB, Ethen MK, Canfield MA, Watkins S, Wantman E, Doody KJ. Sixth grade academic achievement among children conceived with IVF: a population-based study in Texas, USA. J Assist Reprod Genet. 2021 Jun;38(6):1481-1492. doi: 10.1007/s10815-021-02170-9; Matsumoto N, Mitsui T, Kadowaki T, Mitsuhashi T, Hirota T, Masuyama H, Yorifuji T. In vitro fertilization and long-term child health and development: nationwide birth cohort study in Japan. Eur J Pediatr. 2024 Nov 18;184(1):24. doi: 10.1007/s00431-024-05883-y).
Line 267: Should be “sperm cells” or “spermatozoa” instead of “sperms”.
Lines 475-477: The authors should clarify that the cited study is on C. elegans.
Lines 579-580: The sentence needs editing.
Lines 621-622: “Adult pups” (an expression used in the cited source) is an oxymoron; presumably, it refers to adult animals that were subjected to certain influences as fetuses/pups.
Author Response
The presented manuscript is an interesting review summarizing new, often controversial data and concepts. Unfortunately, the authors do not make efforts to present also conflicting data and concepts, including theories that have current and long-lasting status of scientific consensus. In recent decades, the enthusiasm accompanying the fast progress of epigenetics led to the publication of many reports about trans-generational non-genetic inheritance without the needed scrutiny, and a revival of Lamarckian views on heredity. In addition to specific cases where such claims have been explicitly disproved (see references below), the concept suffers from the general logical issue of any theory of inheritance of acquired traits: the hidden assumption that the conditions in which the organism’s ancestors developed and lived have a paramount importance, while the conditions in which the organism itself develops and lives have little importance, or none at all. Because the authors are mostly interested in the long-lasting effects (into adulthood) of factors acting on the early embryo, I do not even understand why they put so much stress on transgenerational epigenetic inheritance – if anything, if such a phenomenon exists, it would determine the fate of embryos by factors that had acted generations ago, and thus diminish the importance of factors acting on the actual embryo here and now, which are the main topic of the review.
Response:
We thank the reviewer for this insightful critique and we agree to include conflicting data and concepts addressed by the reviewer into the revised manuscript. We also wanted to clarify our intention in including transgenerational inheritance as we consider it as an important layer when discussing cellular memory. To align with our review in highlighting the embryonic cells ability to reprogram from environmental exposures and transfer these modulations across cell divisions leaves some consideration to the possibility of transgenerational transmission. As the reviewer kindly pointed out the current consensus on the matter, we have agreed to include those considerations in our revised manuscript.
Lines 21-22: There are yet no compelling reasons to suppose that such transmission exists in humans (or other mammals), see references below.
Response:
We thank the reviewer for the comment and have revised the manuscript to have a more reasonable and considerable statement.
Line 19-24: “Crucially, memory established during early embryogenesis links these epigenetic modifications to adult long-term phenotypes related to metabolic disorders. These modifications—including DNA methylation, histone modifications, and non-coding RNAs—supports the cellular memory transmission remains across cell divisions and in certain organisms, can be transmitted across generations without alterations to the DNA sequence.”
Lines 32, 205-206, 329-332: These statements contradicts the “all-or-none” theory of early embryonic exposure to damaging factors (e.g. Adam MP. The all-or-none phenomenon revisited. Birth Defects Res A Clin Mol Teratol. 2012; 94(8):664-9. doi: 10.1002/bdra.23029), which is the long-lasting scientific consensus and should be referenced even if the authors disagree with it and cite some data supporting the opposite concept.
Response:
We thank the reviewer for this notion and have included to mention the “all-or-none” theory in our revised manuscript along with alteration of statements to allow consideration towards this theory.
Line 35-38: “When we integrate with the classical “all-or-none” theory, where it suggests that these exposures may result in either embryonic death or complete recovery, it could be interpreted that certain sublethal exposures may induce molecular or cellular alterations that allows the embryo to adapt which will persist within the cells across divisions [1].”
Line 265-267: ”Environmental factors present at conception exert profound influences on mammalian development and can alter the long-term physiological health of offspring after adaptative period to ensure survivability [28-32].”
Line 38-39, 60-71, 97-101, 435-436, 645 ff.: Transgenerational epigenetic inheritance is highly controversial, particularly in mammals (e.g. Heard E, Martienssen RA. Transgenerational epigenetic inheritance: myths and mechanisms. Cell. 2014; 157 (1): 95-109. doi: 10.1016/j.cell.2014.02.045; Horsthemke B. A critical view on transgenerational epigenetic inheritance in humans. Nat Commun. 2018; 9(1):2973. doi: 10.1038/s41467-018-05445-5; Bird A. Transgenerational epigenetic inheritance: a critical perspective. Front Epigenet Epigenom. 2024; 2: 1434253. doi: 10.3389/freae.2024.1434253). “In mammals efficient reprogramming occurs in the early embryo and in the germ line... These two rounds of epigenetic erasure leave little chance for inheritance of epigenetic marks, whether programmed, accidental or environmentally induced” (Heard and Martienssen, 2014). Data initially considered as supporting transgenerational epigenetic inheritance have turned out to be based on mutations: “Several studies… have reported the co-segregation of an abnormal DNA methylation pattern… with a rare disease in two or more generations of certain families. In these cases, the abnormal DNA methylation of the gene under investigation was linked to a mutation in a neighboring gene that removed the transcription termination signal... As a consequence, transcription from this gene extended into the gene under investigation, causing abnormal promoter methylation and gene silencing” (Horsthemke, 2018).
Response:
We thank the reviewer for this thoughtful comment. We have taken this into consideration and have framed the revised manuscript to include these conflicting evidence as well as altering our statements to view transgenerational inheritance as either altered epigenetic reprogramming or an inherited modulation in cellular memory response with a bias towards disease susceptibility.
Line 40-47: “Environmental factors present at conception exert profound influences on mammalian development and can alter the long-term physiological health of offspring after adaptative period to ensure survivability [28-32].”
Line 68-85: “The inheritance of these environmentally induced changes, known as transgenerational epigenetic inheritance, is a key area of study but highly debated [6]. Transgenerational studies show that exposure to pollutants, a high-fat diet, or diabetes can propagate re-productive deficits and metabolic dysfunction for two or more generations [7]; this is from “transgenerational epigenetic inheritance”, which refers to the transmission of gene expression changes across generations without altering the DNA sequence, suggesting that DOHaD-related phenotypes can be reproduced through altered developmental processes based on the cellular memory of a single cell, namely the germ cell. These effects are mediated through alterations of the germ cell function, established during critical developmental windows, however, due to the extensive epigenetic reprogram-ming during early embryogenesis these interpretations should be critically viewed as this process exerts certain limitations towards stable transmission of epigenetic markers across generations. Research conducted on paternal diet showed significant influence on off-spring metabolism through germ cells such as embryo and sperm parameters, creating an “inherited metabolic cellular memory” that can affect multiple generations [8]. This can suggest that not only the exposed individual, but also future generations may be affected by ancestral exposures by indirectly modifying the threshold risk towards developing certain metabolic conditions.”
Line 138-142: “These alterations are not only maintained through cell divisions and differentiation but can also be transmitted to the offspring, influencing the disease risk of offsprings from exposed individuals. This can be seen from the studies of historical famines that provide valuable evidence on this transgenerational epigenetic effect of malnutrition [17, 18].”
Line 906-912: “ DNA methylation could be transmitted between generations in certain mammals, however the functional importance and whether it carries information across generations are unclear. For instance, young transposable elements, due to their evolutionary age, may retain DNA methylation levels during early embryonic development, impacting embryonic gene expression [140]. Despite that, other underlying factors such as genetic mutations or indirect mechanisms should be considered in transgenerational transmission.”
Line 1222 ff…
Lines 193-196: The authors should either provide a reference, or frame the statement as a hypothesis still to be tested.
Response:
We thank the reviewer for this thoughtful suggestion and have altered the statement as a hypothesis in the revised manuscript.
Line 245-258: “These epigenomic modifications may not represent transient responses but could instead become stably fixed as cellular memory, persisting across cell divisions and influencing postnatal metabolic regulation and susceptibility to lifestyle-related diseases. This framework suggests a molecular explanation for the central concept of the Developmental Origins of Health and Disease (DOHaD), which posits that environmental conditions during pregnancy critically determine long-term health outcomes.”
Lines 260-261: Other studies do not find significant differences between children conceived naturally and those conceived using assisted reproduction (e.g. Luke B, Brown MB, Ethen MK, Canfield MA, Watkins S, Wantman E, Doody KJ. Sixth grade academic achievement among children conceived with IVF: a population-based study in Texas, USA. J Assist Reprod Genet. 2021 Jun;38(6):1481-1492. doi: 10.1007/s10815-021-02170-9; Matsumoto N, Mitsui T, Kadowaki T, Mitsuhashi T, Hirota T, Masuyama H, Yorifuji T. In vitro fertilization and long-term child health and development: nationwide birth cohort study in Japan. Eur J Pediatr. 2024 Nov 18;184(1):24. doi: 10.1007/s00431-024-05883-y).
Response:
We thank the reviewer for this clarification and have included a thoughtful consideration within out revised manuscript on studies on assisted reproduction offsprings.
Line 389-391: “Consequently, evidence in recent years suggests that offsprings conceived through ART may experience various effects in adulthood [66, 67], although some reports suggest context-dependent differences [68].”
Line 267: Should be “sperm cells” or “spermatozoa” instead of “sperms”.
Response:
We thank the reviewer for this comment and have revised the term to sperm cells.
Line 397: sperm cells
Lines 475-477: The authors should clarify that the cited study is on C. elegans.
Response:
We thank the reviewer for this thoughtful suggestion and have included the species into the revised manuscript.
Line 958-960: “Studies using C. elegans also demonstrated that H3K27me and Polycomb repressive complex 2 (PRC2) each contribute to epigenetically transmitting the memory of repression across generations and during development [144].”
Lines 579-580: The sentence needs editing.
Response:
We thank the reviewer for this comment and have edited the sentence to improve clarity.
Line 1138-1141: “Under various stress conditions, cells may initiate rRNA transcription; however, the subsequent processing of these transcripts often becomes inefficient. In mice, the ribosomal DNA transcription is initiated at the 2-cell stage, while an abrupt shift towards ribosomal DNA replication begins at the 4-cell stage [178, 179].”
Lines 621-622: “Adult pups” (an expression used in the cited source) is an oxymoron; presumably, it refers to adult animals that were subjected to certain influences as fetuses/pups
Response:
We thank the reviewer for this thoughtful clarification and have adjusting the sentence to improve clarity in the revise manuscript.
Line 1197-1201: “A study on the association between maternal hyperglycaemia and postnatal high-fat diet exposure compromises metabolic parameters and hepatic autophagy in female offspring showed reduced phosphorylated AMP-activated protein kinase (p-AMPK) and autophagy-related protein light chain, LC3-II/LC3-I ratio, the presence of steatosis, oxidative stress, and inflammation later on in adulthood.”
Reviewer 2 Report
Comments and Suggestions for Authors A very current and interesting work. It incorporates recent research contributions, reflected in the extensive and recent literature reviewed and included. However, I couldn't find anything regarding the effects of DNA or RNA alteration in embryo cloning by somatic cell transfer, which is a current and widely used technique worldwide in mammals. I hope you can include it.Author Response
A very current and interesting work. It incorporates recent research contributions, reflected in the extensive and recent literature reviewed and included. However, I couldn't find anything regarding the effects of DNA or RNA alteration in embryo cloning by somatic cell transfer, which is a current and widely used technique worldwide in mammals. I hope you can include it.
Response:
We thank the reviewer for the kind word of encouragement and interest in our manuscript. We also wanted to clarify our initial intention for not including somatic cell transfer in embryo cloning as we wanted our review to mainly focus on naturally occurring environmental factors as well as commonly used assisted reproductive technologies. Even though most of the recent studies we included focuses on animal models, we do hope this review can be a baseline reference in the topic of cellular memory in both animals and humans. Despite that, we have agreed to include somatic cell nuclear transfer embryo models as supportive studies on cellular memory mechanisms when applicable.
Line 872-899: “In the context of somatic cell nuclear transfer (SCNT) embryo models, research has found that SCNT-derived embryos are able to maintain the DNA methylation patterns throughout development, which supports the consistent role of DNA methylation in embryonic cellular memory [139].”
Line 916-918: “As reported by Matoba et al., histone methylation can act as a determining factor on embryonic development capabilities presented by observing SCNT embryo models [141].”
Reviewer 3 Report
Comments and Suggestions for Authors
This manuscript provides a comprehensive and up-to-date review on how exposure to adverse nutritional or environmental conditions during early embryonic stages can affect long-term health (the DOHaD concept). The review encompasses well-established aspects, such as epigenetic processes related to DNA methylation, histone modifications, and non-coding RNAs, alongside newer concepts like functional ribosomal DNA (rDNA) memory and nucleolar dynamics. Notably, this work highlights emerging evidence on transgenerational inheritance, including the critical role of the paternal epigenetic contribution via sperm marks, and proposes autophagy modulation during preimplantation as a promising intervention strategy to enhance embryo quality and mitigate the long-term metabolic risks associated with ART
Despite its thoroughness, the document must discuss several critical points regarding the clinical applicability of the findings before it will be published. The main challenge is that the primary findings have been obtained in animal models, and the actual effects in humans remain unknown. Another scientific challenge is establishing a clear causal and functional relationship, given that many reported epigenetic differences (e.g., methylation) do not directly correlate with changes in gene expression—a significant conceptual hurdle. Furthermore, greater clarity is required regarding the precise mechanism of transgenerational DNA methylation inheritance and how paternal epigenetic influence fully translates into offspring phenotypes. For ART, the most crucial area for improvement remains the current inability to isolate the specific impact of each component within the culture medium or procedure. Addressing this key point is essential for optimizing laboratory protocols and enabling precise interventions (such as autophagy modulation or micronutrient control) to effectively prevent the programming of metabolic diseases in the next generation.
Author Response
This manuscript provides a comprehensive and up-to-date review on how exposure to adverse nutritional or environmental conditions during early embryonic stages can affect long-term health (the DOHaD concept). The review encompasses well-established aspects, such as epigenetic processes related to DNA methylation, histone modifications, and non-coding RNAs, alongside newer concepts like functional ribosomal DNA (rDNA) memory and nucleolar dynamics. Notably, this work highlights emerging evidence on transgenerational inheritance, including the critical role of the paternal epigenetic contribution via sperm marks, and proposes autophagy modulation during preimplantation as a promising intervention strategy to enhance embryo quality and mitigate the long-term metabolic risks associated with ART
Despite its thoroughness, the document must discuss several critical points regarding the clinical applicability of the findings before it will be published. The main challenge is that the primary findings have been obtained in animal models, and the actual effects in humans remain unknown. Another scientific challenge is establishing a clear causal and functional relationship, given that many reported epigenetic differences (e.g., methylation) do not directly correlate with changes in gene expression—a significant conceptual hurdle. Furthermore, greater clarity is required regarding the precise mechanism of transgenerational DNA methylation inheritance and how paternal epigenetic influence fully translates into offspring phenotypes. For ART, the most crucial area for improvement remains the current inability to isolate the specific impact of each component within the culture medium or procedure. Addressing this key point is essential for optimizing laboratory protocols and enabling precise interventions (such as autophagy modulation or micronutrient control) to effectively prevent the programming of metabolic diseases in the next generation.
Response:
We thank the reviewer for the thoughtful comments and agree the need for clarification of the challenges mentioned. We have addressed these limitations and challenges in the conclusion section as well as important considerations upon investigating this topic throughout the revised manuscript.
Line 1307 ff.